# Are All Species Created Equal? A Critique of the “Equal Fitness Paradigm”

**DOI:** 10.3390/biology15010094

**Published:** 2026-01-01

**Authors:** Douglas S. Glazier

**Affiliations:** Department of Biology, Juniata College, Huntingdon, PA 16652, USA; glazier@juniata.edu

**Keywords:** adaptiveness, biological scaling, energy, evolution, fitness, information, natural selection, power and efficiency, regulation

## Abstract

All living species are the result of countless generations of “survival of the fittest”. Therefore, one may suppose that all living species, small and large, have attained the same high level of fitness, at least approximately. Indeed, according to the recently proposed “equal fitness paradigm” (EFP), this fitness universally approximates 22.4 kJ/g, which is quantified as the lifetime energetic production of surviving offspring per parental body mass. However, as I argue, the EFP has several problems. They include a questionable measure of fitness and flawed methods used to quantify and compare it among species. Other measures of fitness, which include the timing of reproduction, a critical factor affecting fitness not considered by the EFP, vary considerably with body size and habitat. The EFP also ignores the profound effects of population abundance and geographical range size on the survival and multiplication of species. In addition, if the EFP were true, natural selection at the species level, which depends on fitness variation, would be impossible. Therefore, I advocate a “variable fitness paradigm” (VFP) originally proposed by Charles Darwin and Alfred Wallace. According to the VFP, fitness varies significantly at multiple levels of biological organization, thus permitting selection at all these levels.

## 1. Introduction

In a democracy, it is said that all people are created equal because they have equal rights under the law. However, in nature, is it also true that all organisms (species) are created equal in some sense? According to some religions, the answer is “yes”, because they were created to fit into a perfectly orchestrated divine design. Furthermore, even if one believes that all species evolved naturally rather than being the result of divine creation, one could consider all living species to be equal because they are “equally evolved” ([1], p. 3). Given that living species are the result of countless generations of natural selection rejecting poorly adapted organisms, one might suppose that they have attained equally high adaptive fits to specific environments, or nearly so. Indeed, several ecologists have argued that species coexistence in ecological communities is the result of the evolution of “equalized fitness” achieved in diverse ways by demographic or ecological trade-offs between major components of fitness, including survival versus growth/reproduction or competitive versus colonizing abilities (e.g., [2,3,4,5,6,7,8,9]). This outlook permits one to embrace two apparently contradictory perspectives at the same time: living species are essentially “equal” in terms of their present survival and thus evolutionary success after millions of years of “struggling for existence”, while also being very different in terms of their form, function, life history, and ecology.

This equal fitness perspective has been developed further from an energetic point of view. According to some ecologists, all species, small and large, have evolved an approximately equal energetic fitness, as defined by surviving offspring production during a lifetime [10,11,12,13]. According to this “equal fitness paradigm” (EFP), although species have evolved in diverse ways along a life-history continuum from high rates of reproduction and mortality (and thus short lives) to low rates of reproduction and mortality (and thus long lives), they have all attained equal or nearly equal lifetime offspring production. Although the EFP has been recently promoted by ecologists with little critical discussion so far (but see [14]), it appears to contradict a fundamental assumption of the Darwinian principle of natural selection, i.e., that evolutionary fitness varies substantially among individuals, populations, species, or other units of selection (see [15,16,17] and Section 3). Indeed, selection (sorting) at any hierarchical level cannot occur without variable fitness. Thus, Darwinian evolution, as commonly accepted, is based on a “variable fitness paradigm” (VFP) in contrast to the recently proposed EFP. Is one of these paradigms incorrect or can they both be accommodated within a larger synthetic framework?

The purpose of my commentary is to address this important question, and thereby show that the EFP, as currently developed, has multiple problems conceptually and empirically. I describe these problems in some detail, and in the process provide vindication for the VFP as originated by Charles Darwin [15] and Alfred Wallace [18] and embraced by mainstream evolutionary biology. Although this vindication may seem obvious to many evolutionary biologists, I chiefly direct my analysis toward ecologists and other scientists who may be contemplating using the EFP in their research program. Note that fitness has been defined and estimated in many ways, and the purpose of my article is not to advocate the universal application of any one fitness measure, but to consider multiple measures when evaluating the relative merits of the EFP and VFP (see Appendix A). However, I do advocate that the concept of fitness should be used in operational, non-tautological ways that allow quantitative comparisons among individuals, populations, species and clades. I also argue that at the population/species level the concept of “fitness” should be distinguished from that of “adaptiveness”, as I operationalize quantitatively in Section 6 and Section A.3.

## 2. The “Equal Fitness Paradigm”

The recently proposed “equal fitness paradigm” (EFP) is based on comparing the interspecific body-mass scaling of the mass-specific production rate of surviving offspring (OP) and generation time (G) or lifetime [10,11,12,13]. From these analyses it is claimed that OP and G scale inversely and thus OP x G (or OPG) is “invariant” because it scales with body mass zerometrically (i.e., with a slope of 0), or nearly so. Since OPG is regarded as a useful energy- and time-based measure of fitness, it is claimed that all species have equal fitness, at least approximately, regardless of their body size.

However, the EFP and the analyses used to support it have several problems, which are explained in the next few sections. They include (1) insufficient justification of the fitness measure OPG used to support the EFP, (2) assuming steady-state populations, though most populations and their energy supplies fluctuate, often greatly and to various degrees, (3) assuming the validity of the “rate of living theory”, though it is not generally applicable, (4) assuming that biological time represents an independent 4th dimension in 3/4-power biological scaling, though it scales allometrically (disproportionately) with other spatial dimensions (e.g., body length), (5) ignoring substantial effects of population size and geographical range size on fitness at the population or species level, (6) providing insufficient support for the EFP by using allometric scaling analyses that have multiple conceptual and empirical problems, including that OPG does not scale zerometrically with body mass in birds and mammals (see Section 2.6), contrary to the EFP, (7) not appreciating that a supposed scaling invariance of OPG is not sufficient to support the EFP, (8) underappreciating and not sufficiently explaining the extensive species variation around the body-mass scaling relationships of OP, G and OPG used to support the EFP, (9) not acknowledging that if the EFP were true, species selection, which depends on variable fitness, would not be possible, and (10) insufficient justification for a claimed biophysical basis of the EFP.

### 2.1. EFP Fitness Measure Has Not Been Sufficiently Justified

Proponents of the EFP have not adequately justified why their measure of fitness should be preferred over others that have been proposed, though they acknowledge that there are other fitness indicators [10,11]. Fitness has been defined in multiple ways and has been applied at multiple hierarchical levels of biological organization (see Appendix A). Even if we restrict our definitions to those that are energy- or time-based, as have the proponents of the EFP, there are multiple possibilities. For energy-based fitness, why not use the concepts of “expansive energy” [19] or “reproductive power” [20,21] (see also Section A.2)? For time-based fitness, why not use the concept of “stability” or “persistence” (i.e., relative survival time), as applied to various units of selection from genes to clades and ecological communities (see, e.g., [22,23,24,25,26,27] and Section A.1)? One might suggest that OPG (total offspring production during a generation time or lifetime) is a preferred measure of fitness because it is both energy- and time-based, though this has not been adequately justified/explained by EFP proponents (cf. [11,28]). If so, why not consider alternative fitness measures that are also energy- and time-based, such as OP/G (offspring production rate divided by generation time), OP/L (offspring production rate divided by lifetime) or OPL/G (offspring production over a lifetime divided by generation time)? One may consider these measures better ways to compare fitness among species with different body sizes and life-history durations than OPG. They are analogous to how evolutionary rate has been scaled by dividing evolutionary change (or rate) by generation time [29,30]. OPL/G also resembles the intrinsic rate of increase (*r*), an oft-used measure of fitness (see below). Using OPL/G, *r*, or other fitness indicators reveals that fitness scales strongly with body size, and clearly is not invariant, as further explained in Section 2.6. Note the consistency of the results for each parameter when compared between birds and mammals. Indeed, they are not significantly different from one another (see 95% confidence intervals in Figure 1 and Figure 2 legends in Section 2.6). In addition, the scaling exponents for OPL/G (−0.308 ± 0.066 and −0.314 ± 0.046) have 95% CI that overlap or nearly overlap that for *r_max_* (−0.262 for 44 species of mammals [31]; and −0.26 for 42 species of organisms from viruses to mammals [32].

Consider that the quantity OPG resembles the demographic concept of net reproductive rate (*R*_0_ = “the mean number of female offspring produced per female over her lifetime” [33], p. 56), which has been frequently used as an indicator of fitness [11,23,34,35,36,37,38,39,40,41,42,43]. For a population to persist *R*_0_ must be ≥1. Furthermore, across species of insects, fishes, lizards, birds and mammals, *R*_0_ appears to vary independently of body size or nearly so [44,45,46,47], though this may not be generally true across the tree of life [45]. Similarly, across eukaryotic species, OPG has been calculated to be an average 1 g/g or 22.4 kJ/g over a lifetime regardless of body size [12,13], which are simply biomass and energetic expressions, respectively, of the demographic necessity that *R*_0_ = 1 in steady-state populations. This observation has two implications: (a) biophysical processes or constraints are not needed to explain (purportedly invariant) values of OPG, contrary to claims made by proponents of the EFP [12] (see also Section 2.9); and (b) using OPG or *R*_0_ as a fitness indicator does not adequately account for the effects of reproductive timing on the rate of reproduction or population growth [37,48], i.e., it is not properly scaled to time [41,49]. For example, a deer and deer mouse may have a similar *R*_0_ (or OPG) but since deer mice mature at younger ages and thus have more generations per unit time than deer, their populations can grow significantly faster. According to Roff [36], *r* is a better fitness measure when populations are growing or unstable, whereas *R*_0_ is a better measure when populations are stationary (see also [50]). Hence, the intrinsic rate of increase (*r* = ln *R*_0_/G) is often used as an alternative measure of fitness or population growth capacity (see [16,23,33,34,37,38,39,40,41,48,49,50,51,52,53,54,55,56,57,58,59,60,61,62]). Using G in this way results in a fitness index that is not invariant but rather scales strongly with body size (see, e.g., [31,32,44,55,63,64,65,66,67], Section 2.6, and Section A.3).

### 2.2. The EFP’s Questionable Assumption of Steady-State Populations

The OPG fitness concept of the EFP assumes steady-state populations and constant energy supply. However, most populations fluctuate, often considerably, and generally in a body-size dependent way [55,68,69,70,71]. Energy supply (and thus population carrying capacity) may also vary spatially and temporally [14]. Therefore, during population growth spurts that frequently and episodically occur in small bodied species and in variable environments, birth rates exceed death rates. This leads to selection for higher reproductive power (thus enabling rapid exploitation of temporarily abundant resources and habitats) than observed in large bodied species or in relatively stable environments where birth and death rates are more consistently balanced (following *r*- and *K*-selection theory: ([55,68,72,73,74]; also see [21,75], Section 6, and Section A.3).

### 2.3. The Rate of Living Theory Assumed by the EFP Is Not Generally Applicable

Proponents of the EFP [12] assume that metabolic rate drives the pace of life and death (and thus lifetime), following the rate of living theory [76,77] and the metabolic theory of ecology [78], but this assumption is not generally applicable [79,80,81,82]. More evidence supports the view that the rate of mortality, as mediated by various environmental factors, drives the pace of life, including growth, developmental and reproductive rates, and in some cases, by association, metabolic rate, as well [83].

### 2.4. Biological Time Is Not an Independent 4th Dimension in Biological Scaling

Proponents of the EFP have assumed that biological time represents an independent 4th dimension in quarter-power biological scaling [11,12]. The EFP claims that fitness (OPG) is invariant with respect to body size because its components (OP and G) scale inversely with −1/4 and 1/4 powers, respectively. This quarter-power scaling is assumed to result from biological time (e.g., generation time, G) being an independent 4th dimension commensurate with the three spatial dimensions of biological volumes (see also [45,84,85,86,87,88,89]). However, this view has three problems. First, as described in Section 2.6, the OP and G of birds and mammals do not show quarter-power scaling. Second, G is not an independent 4th dimension, but rather scales allometrically (disproportionately) with the dimension of body length (L) in a variety of organisms, i.e., log–log slopes are typically <1, rather than ≈ 1, as assumed [90]. Third, even if G scaled as L^1^, dimensional scaling analysis [91] indicates that G should scale as M^1/3^, not as M^1/4^ (where M = body mass) as has been assumed [11,12,89]. This claim follows logically from two basic geometric scaling relationships, where G is assumed to scale as L^1^, and M scales as L^3^, assuming that M is proportional to body volume, and body shape is constant.

### 2.5. The Species-Level Fitness Measure of the EFP Ignores Effects of Population and Geographical Range Sizes

The OPG fitness definition of the EFP ignores differences in population size or geographical range size that strongly influence the persistence of populations and species (a major indicator of fitness at these levels: see [92,93,94] and other references cited in Section 3). This is a major omission. Abundant, widespread species can be considered more fit than rare, geographically restricted species. The former often have higher reproductive power (e.g., [21,95]), faster rates of growth and resource acquisition [96], and suffer less extinction relative to allied restricted species (e.g., [97,98,99,100,101,102]), thus being selected for at the species level. They may also have higher speciation rates ([18,92,101,103,104,105,106,107]; but see [108,109,110]), another important driver of species selection [93,111].

In addition, conspecific populations can vary greatly in fitness, as supported by source-sink metapopulation theory [112,113,114]. Small “sink” populations (mortality > natality) are highly vulnerable to extirpation and often can persist only by immigration of individuals from other populations (the so-called “rescue effect”: [115,116]). By contrast, large “source” populations (natality > mortality) that support sink populations with immigrants are more resistant to extirpation and thus are more fit. The relative rates of reproduction and mortality of conspecific populations can vary greatly (see also Section 3); and moreover, these rates do not necessarily exactly balance as assumed by the EFP, because population sizes are also influenced by rates of immigration and emigration not considered by the EFP. Furthermore, immigration can affect the genetic diversity and fitness of individuals and populations [117,118].

### 2.6. Problems with Data and Scaling Analyses Used to Support the EFP

The data and scaling analyses used to support the EFP have several problems (see also [21]). They include (a) conflating generation time and lifetime (they are not the same, contrary to the definition used by [11,12]), (b) the data used for generation time by [10,12] are in many cases unreliable (e.g., many small birds and mammals were reported to have generation times >10 years, which often exceeds their expected lifetimes in nature), (c) the fecundity schedule, an important component of fitness [48,49,119], is ignored, (d) the inverse quarter-power scaling of offspring production rate and generation time across various eukaryotic species reported by [10,12] is based on analyses of datasets that do not contain the same groups of species and higher taxa and thus are not strictly comparable, and (e) analyses of birds and mammals based on the same species do not show inversely related quarter-power scaling, thus invalidating the EFP for these two major taxa [21].

Indeed, the OPG of birds and mammals scales allometrically with body mass (Figure 1A and Figure 2B: based on data collected in [10]), as do alternative fitness measures OP/G, OP/L, and OPL/G (Figure 1B–D and Figure 2B–D; based on data from [10], as well as alternative data assembled in [120] on female age at maturity, an approximate indicator of G, following [49,121], though this measure does not include the effects of iteroparous reproduction). Note that OPG resembles the net reproductive rate (*R*_0_), whereas OPL/G resembles the intrinsic rate of increase (*r*). Ordinary least-squares regression analyses were used because I wanted to determine how variation in each fitness measure is predicted by variation in body mass [122,123]. This method is also appropriate when the Y variable is determined with more error than the X variable [124,125], as is probably the case for my analyses. In addition, I did not use phylogenetically informed analyses because I merely wanted to know whether the slope of a relationship was significantly different from zero. As seen by the obviously negatively trending scatter of points in each graph of Figure 1 and Figure 2, phylogenetic adjustments may alter the exact value of each slope but unlikely its negativity (significantly < 0). Further statistical analyses based on other fitness measures and other taxa are needed to test the generality of the patterns that I have documented.

**Figure 1 biology-15-00094-f001:**
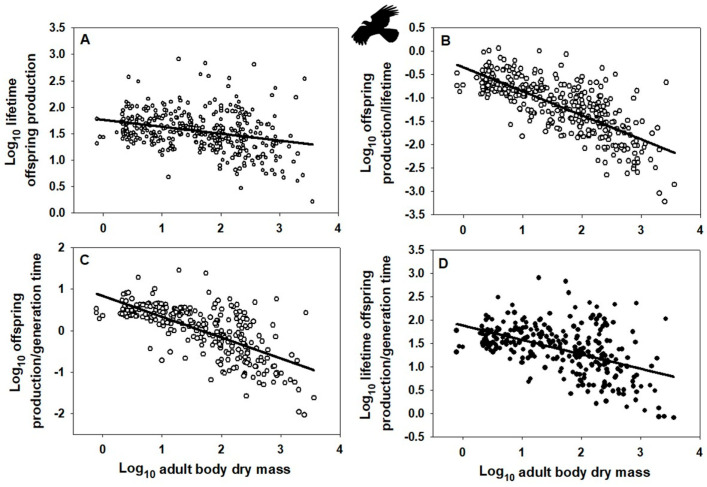
Log–log scaling relationships of various “fitness” indices with adult body dry mass (g) across species of birds (data from [10,120]). (**A**) Mass-specific surviving offspring production (g/g) over a female’s lifetime: Y = 1.757 − 0.131 ± 0.046 (X); r = 0.292; n = 361; *p* < 0.001. (**B**) Mass-specific surviving offspring production rate (g/g/y) divided by a female’s lifetime (yrs): Y = −0.343 − 0.514 ± 0.048 (X); r = 0.744; n = 361; *p* < 0.001. (**C**) Mass-specific surviving offspring production (g/g/y) divided by generation time (approximated by female age at maturity, yrs): Y = 0.833 − 0.501 ± 0.064 (X); r = 0.692; n = 276; *p* < 0.001. (**D**) Mass-specific surviving offspring production (g/g) over a female’s lifetime divided by generation time (female age at maturity, yrs): Y = 1.833 − 0.308 ± 0.066 (X); r = 0.496; n = 276; *p* < 0.001. Error terms for scaling exponents are 95% confidence intervals. Note that the fitness indices in A and D resemble the demographic parameters “net reproductive rate” (R_0_) and “intrinsic rate of increase” (r). Bird silhouette from https://www.phylopic.org/images/, (accessed on 24 December 2025).

**Figure 2 biology-15-00094-f002:**
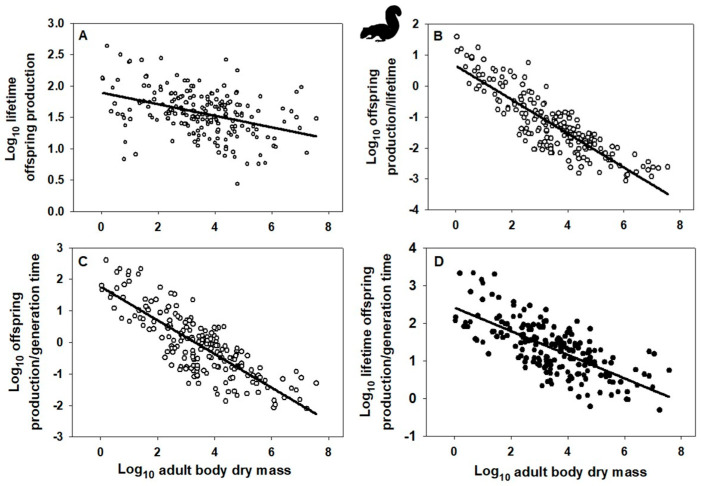
Log–log scaling relationships of various “fitness” indices with adult body dry mass (g) across species of mammals (data from [10,120]). (**A**) Mass-specific surviving offspring production (g/g) over a female’s lifetime: Y = 1.891 − 0.092 ± 0.030 (X); r = 0.395; n = 209; *p* < 0.001. (**B**) Mass-specific surviving offspring production rate (g/g/y) divided by a female’s lifetime (yrs): Y = 0.657 − 0.548 ± 0.046 (X); r = 0.860; n = 209; *p* < 0.001. (**C**) Mass-specific surviving offspring production rate (g/g/y) divided by generation time (approximated by female age at maturity, yrs): Y = 1.771 − 0.535 ± 0.054 (X); r = 0.665; n = 195; *p* < 0.001. (**D**) Mass-specific surviving offspring production (g/g) over a female’s lifetime divided by generation time (female age at maturity, yrs): Y = 2.416 − 0.314 ± 0.046 (X); r = 0.698; n = 195; *p* < 0.001. Error terms for scaling exponents are 95% confidence intervals. Note that the fitness indices in A and D resemble the demographic parameters “net reproductive rate” (R_0_) and “intrinsic rate of increase” (r). Mammal silhouette from https://www.phylopic.org/images/, (accessed on 24 December 2025).

### 2.7. Problems with Basing the EFP on a Supposed Body-Mass Scaling Invariance

Proponents of the EFP focus on body-mass scaling relationships while largely ignoring the substantial influence of other factors. This leads to two major problems as described below.

#### 2.7.1. Body-Mass Independent Variation in Fitness

Allometry-based invariance analyses are inadequate because they ignore variation independent of body size. Even if an allometric invariance is found (scaling slope = 0), much residual variation remains. Indeed, the residual variation for the fitness measure OPG is about two orders of magnitude (100-fold) or more for birds and mammals (Figure 1A and Figure 2A). EFP proponents acknowledge the existence of body-size independent residual variation but regard it as being due to methodological error or other factors of secondary importance [12]. Furthermore, direct relationships between the parameters (e.g., OP and G) comprising a so-called invariant ratio or multiplied quantity (e.g., OPG) may not show a proportional positive/negative 1:1 relationship as assumed [21,90]. In addition, when each parameter is scaled against body size, they may not show exactly inverse relationships as expected for an invariance (including offspring production rate and lifetime or generation time, as assumed by the EFP: see scaling analyses for birds and mammals reported in [21,32]). As a result, the fitness measure (OPG) upon which the EFP is based scales allometrically in relation to body mass (Figure 1A,B) and therefore cannot be considered invariant.

Consider how reproduction and survival often show trade-offs within and among individuals in a population (as reviewed by [36,37,119]; see also data analysis by [126]; and more recent studies by [127,128,129]). The existence of individuals showing such trade-offs implies that reproductive output over a lifetime is less variable than each component parameter examined by itself. However, this pattern does not mean that all individuals have equal or nearly equal fitness. Yet, this appears to be the reasoning analogously used at the species level by proponents of the EFP. Trade-offs between offspring production rate (OP) and survival (lifetime or generation time) occur among species but given that OP over a lifetime may be less variable than each component parameter by itself, does not mean that all species have attained similar fitness. Lifetime reproduction varies considerably among individuals and species (see Section 3). For example, individual variation in growth rate can cause large variation in lifetime reproduction [130]. As noted in Section 2.1, using lifetime reproduction as a fitness indicator also ignores major effects of the rate/timing of reproduction on fitness [37,48]. Species with earlier reproduction may have higher fitness [129,131].

#### 2.7.2. There Are Many Kinds of Scaling Invariances That by Themselves Do Not Provide Sufficient Support for the EFP

The energetic fitness measure of the EFP represents one of several supposedly invariant quantities that have been reported based on allometric analyses (see e.g., [45,82,89,132,133,134,135]). If they are valid (but see Section 2.7.1 and below), one could argue that other invariant quantities could qualify just as well as evidence for equal fitness in species, small and large. For example, the number of breaths or heart beats per lifetime has been reported to be relatively constant across small to large mammals (e.g., [45,136]) and animals generally [82]. If so, does this indicate equal fitness across animal species? I have yet to see a good argument supporting this view. Of course, one may regard these invariants as inappropriate examples because they do not involve reproductive fitness. Wieser [137] showed that the rate of energy expenditure for reproduction scales isometrically or nearly so (slope ~ 1) with litter or clutch mass across animal species, thus resulting in an approximately invariant energy cost of reproduction per unit offspring mass (~250 kJ/kg/d), but this is not surprising given the similar biochemistry underlying biosynthesis in all animals. Moreover, lifetime reproductive effort (a dimensionless quantity analogous to OPG) does not necessarily show zerometric scaling (scaling slope = 0), as predicted by the EFP. Although the scaling slope is apparently not significantly different from 0 in squamates and birds, it is significantly negative in amphibians and mammals [135]. Similarly, Peters [134] examined the scaling of various measures of reproductive effort in mammals, most of which showed a negative relationship with body size (see also the negatively allometric scaling analyses of mass-specific reproductive rates reported by [138]. As another example, across various animal populations, production efficiency (production/assimilation) has been shown to scale with a slope ~ 0 [139,140]. Given that energetic efficiency is often thought to be maximized by natural selection (in fact, Slobodkin [141] proposed this specifically for population production efficiency; see also Section A.2), one could conclude that a scaling invariance for population production efficiency is evidence for equal population-level fitness across animals with different body sizes. If so, we have a quandary because some EFP proponents have argued that natural selection does not maximize efficiency (see e.g., [13,20,28]). However, this quandary is only apparent because none of the so-called invariant quantities that have been reported show that fitness is truly invariant. They merely show zerometric scaling with body mass (slope ≈ 0), while ignoring extensive variation around each scaling line. Furthermore, these and other so-called invariances (e.g., the apparent zerometric scaling of total population energy expenditure with body mass, which was originally reported in mammals and other terrestrial animals and called the “energy equivalence principle” [142,143] and later also observed in some autotrophic organisms ([144]; but see deviant comparative results reported for other taxa: e.g., [145,146,147,148]; and in a controlled experimental system [149]), merely show that multiple traits exhibit either parallel or inverse relationships with body size, or nearly so (see [45,134]). Inferring that zerometrically scaling ratios or multiples of these traits represent equal fitness across body-size classes is problematic because no single preferred fitness measure has been conclusively identified. In addition, correlations between life-history traits (e.g., trade-offs between offspring production and longevity, whether related to body size or not) are “inevitable” because “in an isolated constant-sized population, natality must be balanced by mortality” [150]. Demography of persistent populations prevents rates of growth, reproduction and mortality from varying independently [151]. However, demographic constraints need not entail the attainment of equal fitness by all species. Indeed, differences in population size can have major effects on population or species fitness (see Section 3).

### 2.8. If the EFP Were True, Species-Level Selection Would Not Be Possible

If equal fitness occurred among individuals, populations and species, selection (or sorting) would not be possible at these levels (see also Section A.1). Accordingly, in this sense the EFP is fundamentally anti-Darwinian, even though it was derived based on an energetic definition of Darwinian fitness. If the EFP were true, a lack of species fitness variation would paralyze species selection, thus creating a fundamental problem for current evolutionary theory.

Although proponents of the EFP clearly declare that “at steady state, species are equally fit because they allocate an equal quantity per gram of energy and biomass to surviving offspring” [13], one might loosen this claim by allowing fitness to vary somewhat around a specified central tendency or approximate “invariant” quantity. Indeed, I have carefully used the word “approximately” and similar qualifying words throughout my article to refer to the “nearly” equal fitness claim as presented by [11]. Nevertheless, even in this case, selection would be severely limited by an alledgedly narrow canalization of fitness variation around a reputedly constant mean value (effectively constituting a uniformly restricted “fitness zone”) across species with different body sizes (see also Section 7 and Section 8). Furthermore, EFP proponents claim that their energetic measure of fitness should be regarded as “distinct” from other fitness indicators that are subject to natural selection [10,11]. Indeed, they regard the equalization of their fitness measure across species as being applicable only under steady-state conditions where natural selection is not operating. Thus, they use the term “fitness” in a way that is essentially non-Darwinian and outside mainstream evolutionary thought.

### 2.9. The Biophysical Basis of the EFP Is Questionable

In my opinion, the proposed EFP embodies another problematic attempt to explain evolution simply in terms of universal physical principles or processes (see also [152]). Proponents claim a biophysical basis for the EFP, when none is needed because it is simply an expression of a fundamental demographic constraint (birth and death rates must balance in stable persistent populations; see also Section 2.1) that has been recognized in evolutionary and life-history studies since the 1800s (e.g., [18,45,73,92,150,151,153,154,155]). Other failed attempts to explain evolution in terms of simple physical laws or internal forces include Haacke’s and Nägeli’s orthogenesis, Haeckel’s recapitulation theory, and Kleiber’s 3/4-power law [156,157,158,159,160]. The application of physics to evolutionary theory and biological scaling continues to be a controversial subject demanding more clarity (see e.g., [152,159,161,162,163,164,165,166,167] and Section 4). In any case, the EFP represents another questionable endeavor to impose order on a contingent world, a recurring theme in the history of biology (see e.g., [168,169,170]).

However, I want to make clear that I am not against attempts to find/explain major laws or general trends concerning evolution, which I have acknowledged approvingly or even proposed myself (e.g., the evolution of general allometric and habitat-specific trends in energetic power and efficiency) (see Section 4 and Section 6 and Section A.2 and Section A.3). Rather I am against developing allegedly universal, over-simplistic, physical laws of evolution that ignore the multi-mechanistic basis of evolution and its diverse, opportunistic, and contingent pathways that depend on specific biological and ecological contexts.

## 3. The “Variable Fitness Paradigm”

An alternative view to the EFP is the variable fitness paradigm (VFP) originated by Darwin [15,92,171] and Wallace [18], which continues to be a major conceptual framework used by mainstream evolutionary theory (neo-Darwinism). The VFP recognizes extensive variation in fitness (or its components) at the individual, population and species levels. Note that although EFP proponents claim that fitness equality exists at the species level, they acknowledge that fitness varies among individuals in a population [11]. However, fitness variation at one level may affect fitness variation at other levels. Hence, in this section, I will briefly discuss fitness variation at various hierarchical levels, with a special emphasis on their cross-level interactive effects.

Individual variation (both genotypic and phenotypic) in vital rates, including survival (mortality), growth, and reproductive rates, as well as lifetime reproduction (a commonly used measure of fitness) within populations has been well documented (e.g., [35,42,52,53,92,171,172,173,174,175,176,177,178,179,180,181,182,183,184,185,186,187,188,189,190,191]). Fitness variation in populations is expected to be promoted by sexual reproduction (recombination), which according to theory and experimental evidence facilitates rapid evolutionary adaptation to new or changing environments [192,193,194,195,196,197,198] and furthers population persistence [199]. In addition, fitness and its components (e.g., survival and reproductive success) vary considerably across populations occupying different parts of a species geographical range. Geographical fitness variation in animals and plants may relate to a variety of demographic and abiotic/biotic environmental factors (e.g., [200,201,202,203,204,205,206,207,208]). Fitness and associated traits (e.g., aerobic scope, scope for growth, lifetime reproductive success, population size, and population growth rate) usually decrease in stressful environments (e.g., [209,210,211,212,213,214,215,216,217,218,219,220,221,222,223,224,225,226,227,228,229]). Population fluctuations, which can affect extinction risk (and thus population-level fitness), may also vary considerably across species geographical ranges [230].

Furthermore, extensive variation in persistence (extinction vulnerability) and speciation rate occurs among species, both across the tree of life and within clades of related species [93,109,231,232,233,234,235,236,237,238,239]. Emergent traits at the species level can affect this variation, including population abundance/dynamics and geographical range size ([92,93,94,98,99,100,106,110,233,234,235,239,240,241,242,243,244,245,246,247,248,249,250,251,252,253,254,255,256,257,258,259,260,261]; but see [262]). The wide variation in these species-level traits alone indicates that fitness must vary greatly among species, thus further supporting the VFP. Indeed, the geographical range sizes of species can vary by as much as 12 orders of magnitude [263], and related species-level fitness traits (e.g., rates of speciation and extinction) should thus vary considerably as well.

Individual traits and their variability may also affect emergent species-level fitness because populations and species with more individual variability may persist longer, especially during stressful periods, than those with lower variability ([97,178,264,265,266,267]; but see [100]). Therefore, variable fitness at the individual level can promote selection both within and among populations or species. Abundant, widespread species often (but not always) exhibit higher genetic variation than rarer, more restricted, related species (see e.g., [92,268,269,270,271,272,273,274,275,276,277,278,279]), a trend also documented by general reviews or meta-analyses [277,280,281,282,283]. Furthermore, the population size of both animals and plants is significantly correlated with both genetic variation [277,283,284,285] and reproductive fitness (defined as adult progeny production or population growth rate [286], or as female reproductive output gauged as number of flowers, seeds or fruits [283]). The effectiveness of sexual reproduction in promoting adaptation to changing environments (and thus greater evolutionary persistence) is also higher in larger populations [195,287]. In addition, individual traits such as body size, fecundity, parental care, torpor ability, morphological complexity and niche specialization can affect extinction risk (e.g., [72,231,232,233,234,237,243,247,249,250,251,288,289,290,291,292,293,294,295,296,297,298,299]. Various individual physiological and life-history traits may also affect the growth rate and thus fitness of populations (e.g., [66,300,301,302,303]). Furthermore, the genetic variability of animal populations and thus their prospects for evolution and survival are generally negatively related to body size and longevity [304]. The above analyses connect fitness indicators estimated at the individual and population levels.

Population size or its variability is often correlated with geographical range size, thus connecting these fitness indicators at the population and species (metapopulation) levels [20,69,70,108,240,305,306,307,308,309,310,311,312], as well. Connections between fitness at the individual and species levels are also supported by positive correlations observed in some taxa between geographical range size and reproductive output (e.g., [21,95,307], and references therein) or individual/population growth rate (e.g., [308,313]). Fitness measures may not only be estimated at several hierarchical levels, a “hierarchical expansion” of the theory of natural selection [266,314,315,316], but they may also be linked in multiple ways (see also [25,317]).

Other observations attest to the importance of population size, geographical range size and body size in affecting extinction rates. For example, geographically restricted species tend to occupy relatively stable, predictable, sheltered, or non-seasonal habitats, where extinction risk is reduced, thus revealing species-level selection for population stability (see e.g., [95,242,307,318,319,320,321,322,323,324]). In addition, the maximum body size of animals is negatively correlated with habitat area, because population sizes of large animals are too small for extended persistence in small habitat areas (see [289,325,326,327]). Similarly on continents, large fishes, birds and mammals have relatively large geographical ranges, presumably because those with smaller ranges (and related low population abundances) went extinct [20,108,309]. In addition, theoretical analyses show that time to extinction decreases with increasing population carrying capacity (*K*) and with increasing population growth rate (*r*) under many demographic conditions [328].

Further research is needed to characterize fitness variation taxonomically, allometrically, and ecologically in both space and time. For example, the “fitness landscape” concept [329,330,331,332,333,334] may be used to characterize individual or taxonomic variation in fitness or its components. As a case in point, Beausoleil et al. [335] showed a multi-species pattern of fitness peaks and valleys (based on relative survival) for phenotypic variation in beak morphology among species of Darwin’s finches. Allometric variation can be represented by the body-size scaling of fitness measures such as OP/G (offspring production per generation time) OP/L (offspring production per life time) or OPL/G (offspring production over a lifetime per generation time) (Figure 1B–D and Figure 2B–D) or the intrinsic rate of increase (see Section 2.1, as predicted by [21,68]). One way to consider ecological variation of fitness is to plot a fitness measure along an environmental gradient in space or time [334]. For example, relative reproductive and mortality rates should be expected to vary with resource availability, habitat stability or duration, and stage of ecological succession. Species that occur in relatively unstable, disturbed or seasonal habitats with episodically high resource supplies tend to have higher mass-specific reproductive outputs than related species that occupy relatively stable crowded habitats where resource competition is intense [55,72,73,95,307], see also Section A.3). Shifts in relative allocation of resources to reproduction versus competitive survival (*r*- to *K*-selection) also occur during the ecological or seasonal succession of various kinds of aquatic and terrestrial communities [336,337,338] and with the evolution of increasing body size [339]. All in all, fitness and its components may vary in response to a variety of physical, biological and ecological factors (constraints). Adopting a VFP is not only important for evolutionary theory but also ecological theory (see e.g., [178,265,340,341,342,343,344]). Ecologists who have promoted the view that species coexistence is facilitated by the evolution of equalized species fitness would do well to contemplate the extensive evidence supporting the VFP.

## 4. Equal or Variable Fitness: A Question of Universal Determinism Versus Contextual Contingency

One might argue that the EFP is to be expected because how else does one explain why our biosphere contains a diversity of species rather than only one dominant maximally fit species. Since all living species are the descendants of lineages that have been subjected to countless generations of selection, it might seem reasonable to suppose that they have all attained equal (maximal or nearly maximal) levels of fitness, or nearly so, in various ways (including by demographic trade-offs), thus allowing their present coexistence (see also Section 1). However, coexistence (non-replacement) is not a sufficient indicator of equal fitness, because heterogenous environmental conditions at various spatial and temporal scales (local to global and hours to millennia), the continual occurrence of mutations, sexual recombination and other kinds of “accidents” (random events), and the complexity of organisms, which creates conflicting fitness advantages for various traits, can cause individuals in populations and species in clades to have different abilities to survive and multiply, thus continually generating variable fitness. Stochastic genotypic and phenotypic variation in variable, heterogeneous environments not only fuels selection but also prevents the attainment of equal fitness, which is not as paradoxical as it may seem. Equal (maximal) fitness can only be attained in a constant, homogenous, completely deterministic world (cf. [14]). Indeed, some population geneticists and mathematical biologists have claimed that maximal fitness may often not be attainable, as it strongly depends on contingent conditions (see the varied discussions of [345,346,347]). The stochasticity, variability and heterogeneity of the world continually create diversity, a trend that is so ubiquitous that it has been considered a universal law of evolution [348]. This is ironic because the EFP itself represents an attempt to find a universal law of life based on deterministic physical principles or processes. However, like the biogenetic and 3/4-power laws, the EFP does not represent a universal law (see also Section 2.9). Recapitulation of phylogeny by ontogeny (as dictated by the biogenetic law) is not universal (as evidenced by diverse patterns of heterochrony [157,349,350], nor is the 3/4-power law (as evidenced by diverse biological scaling relationships [83,160,351,352,353,354]). The EFP is not universal either, because of ubiquitous variation in fitness at multiple levels of biological organization (see also Section 3 and Section 6 and Section A.3).

The VFP may even apply to the structure and function of ecological systems, which may vary in terms of their long-term stability, thus subjecting them to persistence selection [24,25,27,355,356]. Examples include various food-web patterns [357], biochemical cycles [26,358,359,360] and other functional networks of multi-species communities and ecosystems [361,362]. The VFP coupled with persistence (stability) selection may also have played an important role in the origin of life itself [22,363,364,365,366,367]. This makes sense because stability selection undoubtedly has played an important role in the evolution of chemical systems. As Calow ([368], p. 5) remarked: “Only stable chemicals persist”.

Accepting the widespread applicability of the VFP not only represents a vindication of the view of life originated by Darwin and Wallace but also represents an important step toward rejecting overly simplistic, deterministic, biophysical, “Newtonian” approaches to the study of evolution in favor of multi-faceted, contextual, ecological “Darwinian” approaches (as has also occurred in the field of biological scaling [160]). However, this does not mean that life does not obey fundamental physical principles. As recognized by Darwin [92] and other evolutionary biologists, life has evolved in a physical world and is thus constrained by physical principles, though the nature and extent of influence of these constraints is not wholly clear or commonly accepted (see e.g., [21,159,160,166,167,369,370,371,372,373,374,375,376,377,378,379,380]). Physical (energetic) constraints are manifested in part by the covariation between energetic power and efficiency observed in living systems, as discussed in [21], Section 6, and Section A.3. However, despite these inescapable constraints, life has evolved a remarkable diversity of form, function and fitness, which has been made possible by the coordinated acquisition and use of both energy (resources) and information, the subject of Section 5.

## 5. Toward Evolutionary Theory That Integrates the Acquisition and Use of Both Energy and Information

I contend that fully understanding evolutionary fitness and how it varies among individuals, populations, species and clades requires that we view fitness not only in terms of “energy” or “genes”, but both. Organisms have evolved to be “well-informed resource users” [159,381]. Their fitness and adaptiveness are not simply passive results of energetic/physical constraints but are flexible, well-informed (actively regulated) responses to diverse environments. Genes and various biological regulatory systems play important roles in controlling the expression of fitness in various ecological contexts. We need to unite the energy and information paradigms discussed by Van Valen [382]. I believe that this will be important not only for evolutionary biology specifically, but also biology generally. Eldredge [383] similarly suggested that genetic and economic (energetic) views of life should be synthesized to obtain a comprehensive understanding of evolution. However, I argue that we should go beyond Eldredge’s view that “Genetic information acts as a ledger book, a record of the status of biological systems up to a given moment” (p. 352, see also [25]) to consider how genetic information and other information-based regulatory systems determine how and to what extent organisms acquire and use energy (resources) in the face of various environmental challenges [80,381]. Interactions between the acquisition/use of energy and information are important causes of fitness variation and in turn the operation of natural selection in the living world (see also [62,384,385,386,387,388]). The evolution of fitness is not constrained merely in a “passive” way by biophysical (energetic) factors (as assumed by EFP proponents), but rather is mediated by “active”, highly flexible biological regulation based on information acquired about the external world (cf. [389]). Studies of how regulatory networks determine the effects of environmental stress on the growth and fitness of organisms and populations illustrate this fact very well (e.g., [224,225,390,391,392,393,394,395,396,397]). A related viewpoint is that “agency” (i.e., goal-directed power) can significantly alter the availability of energy for fitness-related activities [14].

An analogous situation exists for many ontogenetic growth models, which assume that growth (an important component of fitness) is merely the passive result of energy acquisition exceeding energy costs for maintenance, such that when they become equal growth stops (e.g., [398,399,400]). These models ignore the influence of information-based regulatory systems, which can cause significant variation in growth rates and the energetic processes supporting them ([80,393,396]), as revealed by responses of prey growth (and metabolic) rates to different predation regimes (e.g., [401,402]).

Given the importance of both energy (resources) and information (about resources and environmental hazards) for evolutionary fitness, I recommend future studies that examine the spatial and temporal variation of fitness in relation to energy, nutrient, and fear landscapes and the abilities of organisms to gain useful information about them. Energy landscapes concern variable energy costs of movement across landscapes [403,404,405] that can impact foraging for survival, growth, reproduction, and ultimately fitness [406]. Nutrient landscapes concern variable nutrient availability across landscapes [407,408,409] that influence the ability to survive, grow and reproduce, and ultimately fitness [410,411]. Landscapes of fear concern variability of predation risk across landscapes [405,412,413] that can affect survival and foraging needed for growth and reproduction, and ultimately fitness [405,414,415]. Temporal variation in energy costs, nutrient availability and predation risk can also cause variation in fitness over time [14,407,416,417,418].

## 6. Species Diversity and Coexistence Are Enabled by Size and Habitat Spectra of Fitness (Power) and Adaptiveness (Efficiency)

How do so many living species coexist, if they have not attained a similar level of fitness and thus evolutionary competitiveness, as proposed by the EFP and other ecological species coexistence models? If fitness (as defined by “reproductive power” or the rate of offspring production or population growth rate: see Section 2.1 and Section A.2 and Section A.3) is highest in the smallest organisms and lowest in the largest organisms, how were large organisms able to evolve and persist without eventually being replaced by smaller organisms. Why isn’t the modern world dominated only by high-fitness microbes with high reproductive power? This may be because large organisms have compensated for their lower reproductive power by having higher survival ability that is associated with higher adaptiveness and efficiency of resource acquisition. Many large organisms eat smaller organisms or outcompete them for shared resources because of their higher efficiency of resource acquisition. The living world can be envisioned as a size-spectrum from high fitness and reproductive power at the small end to high survival, adaptiveness, and efficiency of resource acquisition at the large end (Figure 3; see [21,339] for supporting details).

According to Glazier [21] and *r*- and *K*- selection theory (as interpreted by [55,68]), species follow a similar trend from high fitness (power) to high adaptiveness (efficiency) along a habitat-spectrum from low to high stability (Figure 3). These hypothetical spectra depend on defining fitness and adaptiveness energetically (see Section A.2) and as reproductive power versus efficiency of resource acquisition, respectively (see Section A.3 for details). Although Glazier [21,83,339] provides empirical support for both spectra, they require further testing. Furthermore, these indicators of fitness and adaptiveness are proposed as useful comparative tools at the population and species levels, and not as universally applicable measures at all levels of biological organization.

Intriguingly, my conceptual scheme partially parallels the EFP and other ecological models that explain the coexistence of species as the result of their having evolved different life-history strategies along a continuum involving a fundamental trade-off between reproduction and survival. However, unlike the EFP and some ecological species coexistence models that are based on species fitness equalization, my scheme is based on a fundamental trade-off between fitness (power) and adaptiveness (efficiency). My view has the advantage of helping to explain the diversity and coexistence of species, while working within the VFP, which is required for natural selection to operate, as commonly accepted by evolutionary biologists. It also makes sense in terms of a recently proposed “mortality theory of ecology” that helps explain diverse biological scaling patterns [83,90,339,419]. Small organisms, or those living in relatively unstable habitats, suffer relatively high rates of mortality that favor high rates (power) of compensatory reproduction and thus fitness, as mediated by natural selection and relatively high resource availability per capita (due to populations being frequently below their carrying capacity: see Figure 3). By contrast, large organisms, or those living in relatively stable habitats, experience lower rates of mortality that enable relatively high survival (longevity) and the evolution of relatively high adaptiveness (as indicated by higher efficiency of resource acquisition), which is advantageous because of relatively high intraspecific competition and low resource availability per capita (due to populations being frequently at or near their carrying capacity; see Figure 3, Section A.3, and [83,339]).

## 7. Prospectus for Assessing the Equality/Variability of Species Fitness and Adaptiveness

In this section, I outline some ways that the equality/variability of species fitness and adaptiveness should be assessed, thus further addressing the major question asked in my commentary: “Are all species created equal?”. In doing so, the EFP may serve as a useful null hypothesis [14]. (1) Statistical tests should use a variety of operational (quantifiable) measures of species fitness and adaptiveness to compare among species, not merely the one limited measure used to support the EFP (cf. [14]). (2) Tests should be performed at various spatial and temporal scales, from local communities to the biosphere and over various geological time periods (cf. [14]). The EFP has been formulated based only on existing species that have been sampled indiscriminately from across the globe and thus ignores how species fitness or adaptiveness may change across different kinds of habitats and geographical regions and over evolutionary time. The EFP also posits that species have equal fitness “at all scales, from local populations and communities to the global biota” ([11], p. 1272), a bold claim that should be tested. (3) Comparisons of species fitness and adaptiveness should be examined among habitats with different levels of stability (e.g., seasonal versus non-seasonal; and disturbed versus undisturbed) to determine whether species fitness equalization more likely occurs in stable habitats, thus following the assumption of the EFP that species have steady-state populations. (4) Levels of species fitness and adaptiveness should be assessed across landscapes varying in energy/nutrient availability and predatory risk all of which can affect survival, reproduction, and rates/efficiencies of energy/nutrient uptake/use that contribute to fitness and adaptiveness (see also Section 5). (5) Variability of species fitness and adaptiveness should be compared among different phylogenetic groups both with respect to body size (as I have done for birds and mammals: see Section 2.6) and independently of body size. Species fitness may vary greatly, independently of body size, and thus estimates of species fitness equality should not be based merely on whether a species fitness measure is “invariant” in an allometric analysis (i.e., shows a log–log slope ≈ 0) (see also Section 2.7). Related studies may include assessing how narrow the residual variation of species fitness around an allometric regression line should be to provide support for the EFP versus VFP, and whether the components of a fitness measure (e.g., offspring production rate and lifetime) covary with negative isometry (log–log slope ≈ −1), as predicted by the EFP. (6) Following recommendation 5, it may also be useful to consider whether an intermediate view between the EFP and VFP best applies in some cases. Perhaps fitness varies but only within specific constraints or boundaries: if so, this could be called the constrained fitness paradigm (CFP). Methods are needed to distinguish between the EFP, CFP and VFP.

## 8. Conclusions

I have contrasted the recently proposed EFP with the VFP originated by Darwin and Wallace. The EFP claims that since individuals of all species produce about the same amount of offspring biomass (or energy equivalent) per adult mass per generation (lifetime), they all have the same fitness (defined as OPG), at least approximately. However, the EFP has several conceptual and empirical problems, the most important of which are provided here as conclusions (see Section 2 for details). First, the measure of fitness (OPG) proposed is not justified. Why it should be preferred over other possible fitness measures (indicators) is not explained adequately (see also Section A.1). Second, the EFP assumes that species populations are generally stable, or nearly so, such that rates of reproduction and mortality balance (which underlies the claimed constancy of OPG among species), but most natural populations fluctuate greatly, especially for relatively small organisms, which according to life-history theory can have great consequences for the evolution of fitness (reproductive success). Third, despite being based on interspecific analyses, the EFP ignores the profound effects of population abundance and geographical range size on fitness at the population and species levels. Fourth, the EFP is based on comparing broad interspecific scaling analyses of eukaryotic offspring production and generation time, which is problematic because these analyses are based on misleading and frequently unrealistic estimates of generation time, as well as comparisons of scaling relationships that contain many non-overlapping species and taxa. Therefore, these scaling analyses are unreliable and non-comparable. Fifth, scaling relationships for mass-specific offspring production rate and generation time (using data collected by proponents of the EFP) analyzed for the same sets of species in two major taxa of animals (birds and mammals) are not exactly inverse, thus invalidating the EFP in these taxa. Sixth, the EFP is based on a purported “scaling invariance” that not only does not exist in birds and mammals but also ignores extensive variation unrelated to body size. Seventh, mass-specific offspring-production rate and generation time in birds and mammals are not related in a negatively isometric way (slope ≠ −1), as predicted by the EFP. Thus, OPG is not constant in either birds or mammals, but varies significantly with body size, as do alternative fitness measures (e.g., the intrinsic rate of increase, r, OP/G, OP/L, and OPL/G: see Figure 1 and Figure 2; and Section 2.1 and Section 2.6).

By contrast, the VFP recognizes extensive fitness variation at the individual, population and species levels. This view is vindicated by multiple energetic, allometric, comparative, and hierarchical analyses, as discussed in Section 3. First, I review numerous studies that show that fitness, defined as differential multiplication and/or persistence, varies enormously at the individual, population, species and clade levels. At the population and species levels, important fitness correlates include population abundance and geographical range size. In general, abundant, widespread species tend to persist longer than do rare, geographically restricted species. Second, fitness variation is essential for continuing multilevel selection at the individual, population, species and clade levels, which would not be possible at the species level if the EFP were true. The existence of equal fitness at specific hierarchical levels would cause evolutionary change via natural selection to be at a standstill at those levels. Third, fitness variation at the individual level may affect fitness variation and resulting selection at higher levels. For example, populations or species with relatively high genetic variation among individuals may persist longer, on average, than those with low variation. In addition, clades containing abundant widespread species may persist longer, on average, than those with rare restricted species. Fourth, individual traits such as body size and reproductive output (which relate to population abundance and geographical range size in various taxa) may affect the persistence (and thus fitness) of populations and species. Often species with smaller body sizes and higher reproductive rates persist longer than related species with larger body sizes and lower reproductive rates. Fifth, I recommend that “fitness” and “adaptiveness” at the levels of populations and species be distinguished as productive “power” versus “efficiency” of resource (energy) acquisition for production (see also [21], Section 6, and Section A.3). This distinction reconciles two largely alternative approaches that have long been used to characterize evolutionary fitness and adaptation (often regarded as equivalent) as either energetic power or efficiency (see Section A.2). Distinguishing them in this energetic way allows one to use the concepts of fitness and adaptiveness operationally and non-tautologically so that they can be compared among species with different biological and ecological characteristics. Sixth, following the above distinction, it appears that fitness (productive power) and adaptiveness (efficiency of energy acquisition for production) decrease and increase, respectively, in species with larger body size and/or that use resources and habitats that are more stable (see Figure 3, Section 6, Section A.3, and [21]). This distinction also helps explain the coexistence of so many species, small and large, in the living world, within the conceptual framework of the VFP. In Section 7, I describe some ways of further assessing the relative variability of species fitness and adaptiveness, and thus the relative validity of the EFP, VFP and a possible intermediate view that I have called the “constrained fitness paradigm” (CFP).

Logic and currently available evidence strongly favor the VFP over the EFP. Consider that the EFP uses selected aspects of the life-histories of currently existing species to formulate a measure of species-level fitness (OPG) and thus represents a time-limited snapshot view of species survival. An implication of the EFP, which is not articulated by its proponents, is that it regards the successful current survival of existing species as being due to their equally maximized OPG attained because of countless past episodes of selection. As such, the EFP represents a “present existence” view of species fitness. In this sense, it is an “equal existence paradigm” that is a truism with little value for predicting future survival or reproductive success. By contrast, the VFP takes a more time-expansive (prospective) view of species fitness as a propensity for “future persistence” (see also [14,23,420,421]). By focusing on how currently living species trade off offspring production with mortality rate (as indicated by generation time or lifetime), a demographic necessity for relatively stable populations, the EFP gives incomplete information about the likely future survival and reproductive success of a species. For example, compare two species with purportedly equal fitness (as defined by OPG) but one is abundant and widespread whereas the other is rare and restricted. The EFP gives us no clue about the likely greater future persistence of the widespread versus restricted species, even though they may have equal OPG. The large amount of variation in fitness (and fitness-related traits), as related to likely future persistence and multiplication, that has been observed at multiple levels of biological organization firmly supports the VFP over the EFP (see Section 3 and Section 4). Therefore, the viewpoint of Darwin and Wallace regarding the importance of fitness variation in evolution is vindicated, thus upholding a continuing Darwinian Revolution in scientific thought.

## Figures and Tables

**Figure 3 biology-15-00094-f003:**
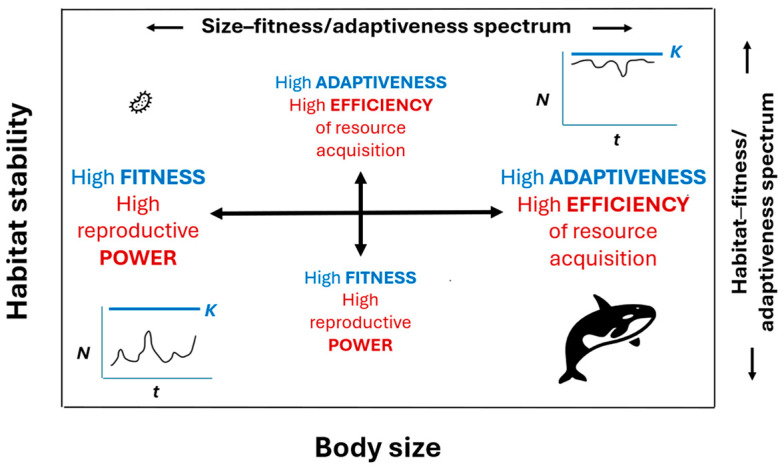
Schematic representation of the hypothesis that fitness (as indicated by reproductive power) decreases, whereas adaptiveness (as indicated by efficiency of resource acquisition) increases with increasing body size (from microbes to whales) and habitat stability. Both trends are linked to decreasing mortality rates and increasing intraspecific competition (and thus decreasing resource availability per capita) that relate to more stable populations whose size (*N*) over time (*t*) is closer to their carrying capacity (*K*). For more explanation, see the text (Section 6 and Section A.3) and [21,83,339]).

## Data Availability

All data used in this article can be found in the cited references.

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
