# Peer review of "Biology2026, 15(1), 94;https://doi.org/10.3390/biology15010094"

_biology, 2026, doi:10.3390/biology15010094_

Round 1

Reviewer 1 Report

Comments and Suggestions for Authors

This is a robust and compelling critique of the EFP.  Remarkably, its points are almost entirely different from those we raised in our critique (Vermeij. et al., 2025).  I rather wish Glazier would have engaged more with our critique, but this is easily fixed.  I might disagree a bit about attempts to wrestle major laws about evolution, a project Glazier disapproves of I think it is worthy goal, as indicated in my book The Evolution of Power.  The big takeaway for me is the futility of the entire concept of fitness.  It is, as Glazier points out, too malleable, too imprecise, and frankly not very useful when gauging the success or organisms at all scales of space and. time.   We intimated this in our commentary, but Glazier makes an even stronger case for the abandonment of fitness.    A few points.  I would like some shortening of this paper; there is a fair bit of repetition.  I also recommend avoiding abbreviations as much as possible; it's always hard to bear in mind what they all mean.  Basically, however, this is a very thoughtful and important critique of an obviously flawed paradigm.

Author Response

Thank you for your positive assessment of my manuscript. 

Although the useful perspective article by Vermeij et al. (2025) appeared after I had nearly completed writing my article, I had cited it 8 times in my originally submitted manuscript.  In my revised manuscript, the number of citations has increased to 12.  I largely agree with this article, but I do not believe that the concept of “fitness” should be abandoned, as Vermeij believes, but rather that it should be used in operational, non-tautological ways that allow quantitative comparisons among individuals, populations, species and clades.  I also believe that at the species level the concept of “fitness” should be distinguished from that of “adaptiveness”, as I argue (and operationalize quantitatively) in my manuscript and in a previous publication where the allometric analyses upon which the “equal fitness paradigm” is based are briefly criticized (Glazier 2024).

Please also note that I am not against “attempts to wrestle major laws about evolution” or to document/explain general trends, which I have acknowledged by citing the work of Lotka, Vermeij, McShea & Brandon, and others, and also proposed myself (e.g., allometric and habitat-related trends in power and efficiency), but rather I am against developing allegedly universal, over-simplistic, physical laws of evolution that ignore the multi-mechanistic basis of evolution and its opportunistic, contingent pathways that depend on specific biological and ecological contexts.   I have clarified this point in my manuscript.

Since the reviewer does not give specific examples of where my manuscript is repetitive, I am not sure how to deal with this perceived problem. Nevertheless, I have carefully re-read my manuscript and attempted to remove any unnecessary repetition that I could find. Note that the material that I cover is interconnected so I often cross-reference material from different sections of my manuscript.  I hope that this practice is not misconstrued as being “repetitive”. I also apologize for using abbreviations throughout my manuscript, but in my defense, doing so reduces repetitive wording, and the abbreviations are clearly defined at first use and in a list of abbreviations at the end of my paper. 

Literature cited:

Glazier, D. S. (2024). Power and efficiency in living systems. Sci6(2), 28.

Vermeij, G. J., Grosberg, R. K., & Roopnarine, P. D. (2025). Energetics and evolutionary fitness. Proceedings of the National Academy of Sciences122(21), e2423684122.

Reviewer 2 Report

Comments and Suggestions for Authors

The manuscript offers an energetic, conceptually ambitious critique of the “Equal Fitness Paradigm” (EFP) and argues, largely persuasively, that substantial variation in fitness—across individuals, populations, and lineages—is indispensable for understanding adaptation, diversification, and Darwinian selection. The author’s central organizing move is to contrast an alleged EFP claim of near invariance in lifetime reproductive output (or lifetime energy production per gram) with a proposed “Variable Fitness Paradigm” (VFP) in which fitness differences are pervasive and scale-dependent. This is a timely intervention in an active debate about whether macroecological regularities can be elevated to “universal rules” without erasing the contingent, context-dependent processes emphasized by evolutionary theory.

The paper is generally well-structured and readable, engaging a broad literature, and usefully highlights conceptual pitfalls that arise when a demographic or energetic accounting metric is treated as a general-purpose proxy for evolutionary fitness, especially when reproductive timing, non-equilibrium dynamics, abundance, and persistence are not explicitly considered. In that sense, the manuscript succeeds in redirecting attention to the biological meaning of the quantities being compared and the interpretive leap from scaling relationships to evolutionary claims.

At the same time, the paper’s impact is constrained by three issues that should be addressed to make the critique as rigorous and fair as possible. First, the framing of the EFP at several points risks treating the strongest possible version of the opposing view—namely that fitness is literally equal and that selection would therefore be impossible—as the paradigm’s intended meaning. That inference follows if “equal fitness” is construed as the absence of meaningful variance, but many advocates of EFP-like invariance interpret it instead as a long-run, equilibrium-like expectation (or null model) compatible with substantial short-term and context-dependent variation. A more academically robust critique would explicitly articulate this moderate interpretation and then explain why it is still insufficient: for example, because “approximate invariance” in a composite life-history measure may reflect demographic constraints on persistence (e.g., the tendency of R₀ toward unity in long-lived lineages), because it does not track the temporal component of fitness that selection responds to, or because it conflates levels of biological organization.

Relatedly, the manuscript would benefit from a clearer separation of individual-level fitness, population growth dynamics, and species-level success (persistence, range expansion, extinction risk). The current argument often moves between these levels in ways that are intuitively plausible but not always explicit, leaving readers room to object that EFP is being rejected partly because it does not address questions outside its intended scope.

The empirical and mathematical critique—although potentially important—needs greater transparency and, in places, fuller demonstration. The re-analysis of scaling relationships presented in the figures is central to the paper’s case that OPG (or its energetic analogue) is not invariant across major taxa, yet the statistical workflow is not described in enough detail to allow confident evaluation. A reader needs to know precisely which datasets were used, the sample sizes and inclusion criteria, how generation time was defined or approximated, which regression framework was chosen (and why), and whether phylogenetic non-independence and measurement error were addressed. These details matter because conclusions about scaling exponents can be sensitive to model choice (e.g., OLS vs. SMA), taxonomic breadth, and the treatment of shared ancestry. The manuscript also raises a substantive concern about the unreliability of generation-time estimates in some EFP studies, but then relies on “age at maturity” as a proxy (Figure 1 and 2); this is reasonable as a pragmatic choice, yet it is not neutral, especially for iteroparous species where generation time depends strongly on adult survival and the distribution of reproduction across ages. The critique of others’ operationalizations will be more persuasive if the paper explicitly acknowledges the limitations of its own proxy and either provides a sensitivity analysis (showing how conclusions change under alternative plausible definitions) or explains why any bias introduced would not alter the direction of the main findings.

The theoretical challenge to quarter-power scaling requires strengthening. The manuscript’s dimensional-analysis claim that certain time variables should scale with body mass as M˄1/3 rather than M˄1/4 is presented as a key objection to the metabolic theory tradition, but it is asserted more than it is derived. Given how foundational quarter-power ideas are in this literature, readers will expect either (i) a concise derivation with explicit assumptions, or (ii) a clear citation to a published physical–biological model that yields the stated exponent and explains when and why it should supersede quarter-power expectations. Without that, the argument risks being perceived as an interesting intuition rather than a compelling alternative framework.

The VFP proposal is conceptually attractive, but it would be considerably strengthened by being made more operational: the manuscript should articulate a small set of distinct, testable predictions that would differentiate VFP from an EFP-style null expectation, specify the kinds of datasets needed, and state what empirical outcomes would count against the proposed spectra of power/efficiency and the multi-level fitness-variation thesis. Doing so would transform VFP from a largely programmatic stance into a productive research agenda.

Overall, the manuscript advances a valuable and broadly correct reminder that evolutionary explanation depends on meaningful variation in fitness and on careful attention to what particular metrics can and cannot represent. With revisions that (i) engage the most defensible interpretation of EFP rather than its most literal reading, (ii) provide methodological and statistical transparency sufficient for replication and confident interpretation, (iii) substantiate the dimensional-analysis claims with explicit derivations or robust citations, and (iv) sharpen the VFP into a set of falsifiable predictions, the paper would become a more balanced and academically durable contribution to this debate.

Author Response

Thank you for your positive assessment of my manuscript and your many constructive and thoughtful comments.  Below I address the four major issues discussed by the reviewer.

Issue 1: The reviewer suggests that I have not presented the EFP fairly by saying its proponents claim that all species have the same fitness “literally”, when they reputedly acknowledge some variation.  However, note that in several places in their publications, they clearly state that the EFP specifies that all species have equal fitness.  For example, Brown et al. (2024) state that “at steady state, species are equally fit because they allocate an equal quantity per gram of energy and biomass to surviving offspring” (my italics added). However, Burger et al. (2021) do state that according to the EFP, fitness “is nearly equal across species in ecological assemblages” (my italics added).   Therefore, I have been careful to point out that the EFP posits that fitness is equal among species only approximately (as stated repeatedly throughout my manuscript).  I have also added further discussion about this issue in sections 2.8, 7 & 8.  

The reviewer is also worried that my discussion of fitness variation at various levels of biological organization may confuse readers because the EFP only deals with fitness at the species level.  However, my discussion of fitness variation at multiple hierarchical levels is mainly discussed with respect to an alternative view, the “variable fitness paradigm”.  When discussing the EFP, I always specify that fitness is being considered at the species level.  Furthermore, I feel that it is important to discuss fitness variation at various hierarchical levels and their interactive effects, because EFP proponents distinguish fitness variation at the individual level from equal fitness at the species level.  However, fitness not only varies at both the individual and species levels, but also the former may affect the latter (see added material in section 3).

Issue 2: I agree that some specific aspects of my data analysis in section 2.6 should be articulated.  Therefore, I have added the following material: “Ordinary least-squares regression analyses were used because I wanted to determine how variation in each fitness measure is predicted by variation in body mass [122,123]. This method is also appropriate when the Y variable is determined with more error than the X variable [124,125], as is probably the case for my analyses. In addition, I did not use phylogenetically informed analyses because I merely wanted to know whether the slope of a relationship was significantly different from zero. As seen by the obviously negatively trending scatter of points in each graph of Figures 1 and 2, phylogenetic adjustments may alter the exact value of each slope but unlikely its negativity (significantly < 0). Further statistical analyses based on other fitness measures and other taxa are needed to test the generality of the patterns that I have documented.”  Note that information on sample sizes and data sources is already provided in section 2.6, including the legends of Figures 1 and 2. Further details about data selection can be found in the sources, which are compilations of data collected by the proponents of the EFP and another widely used life-history data source (Myhrvold et al. 2015).  I have also qualified my description of age at maturity as an indicator of generation time, as requested by the reviewer.

Issue 3: Apparently, the reviewer has misconstrued my argument in section 2.4.  First, this section is not intended to challenge quarter-power scaling in general (which I have done elsewhere: see e.g., Glazier 2005, 2022), but rather to criticize the view that quarter-power scaling can be explained by simply invoking time as a 4th dimension.   I provide three reasons (not just one as stated by the reviewer) for why this view is problematic.  The reason that the reviewer focusses on is not a mere “assertion” but follows logically from the basic geometric assumptions that G scales as L1, and M scales as L3, as I have now clarified in my manuscript

Issue 4: The reviewer recommends that I make my commentary more “operational” by specifying methods of testing the VFP versus EFP.  Toward this end, I have now added a new section 7 that describes “some ways that the variability of species fitness and adaptiveness should be assessed”, thereby distinguishing the VFP from the EFP and a possible intermediate view called the “constrained fitness paradigm” (CFP).  Also, please note that throughout my manuscript I advocate operational, quantifiable measures of fitness that will facilitate comparative tests of the VFP and EFP.  In addition, section 2.6 serves as an example of how the EFP can be tested. However, much remains to be done to develop specific methods that enable the VFP, CFP and EFP to be distinguished.   

Literature cited:

Brown, J. H., Hou, C., Hall, C. A., & Burger, J. R. (2024). Life, death and energy: What does nature select? Ecology Letters27(10), e14517.

Burger, J.R., Hou, C., Hall, C.A.S., & Brown, J. H. (2021). Universal rules of life: metabolic rates, biological times and the equal fitness paradigm. Ecology Letters24(6), 1262-1281.

Glazier, D. S. (2005). Beyond the ‘3/4-power law’: variation in the intra-and interspecific scaling of metabolic rate in animals. Biological reviews80(4), 611-662.

Glazier, D. S. (2022). Variable metabolic scaling breaks the law: from ‘Newtonian’ to ‘Darwinian’ approaches. Proceedings of the Royal Society B: Biological Sciences 289(1985), 20221605.

Myhrvold, N.P.; Baldridge, E.; Chan, B.; Sivam, D.; Freeman, D.L.; Ernest, S.M. (2015). An amniote life‐history database to perform comparative analyses with birds, mammals, and reptiles: Ecological Archives, E096‐269. Ecology, 96, 3109-3109.

Round 2

Reviewer 2 Report

Comments and Suggestions for Authors

Many of the clarifications and additions have strengthened the manuscript, particularly with respect to methodological transparency and the articulation of the Variable Fitness Paradigm. While some conceptual differences regarding the framing and scope of the Equal Fitness Paradigm remain, these largely reflect interpretive perspectives rather than correctable flaws. Given the manuscript’s nature as a Commentary, I view these differences as constructive and conducive to further scholarly discussion.